# Non-Volatile Transistor Memory with a Polypeptide Dielectric

**DOI:** 10.3390/molecules25030499

**Published:** 2020-01-23

**Authors:** Lijuan Liang, Wenjuan He, Rong Cao, Xianfu Wei, Sei Uemura, Toshihide Kamata, Kazuki Nakamura, Changshuai Ding, Xuying Liu, Norihisa Kobayashi

**Affiliations:** 1Beijing Institute of Graphic Communication, Beijing 102600, China; h18736293925@hotmail.com (W.H.); c17812061939@outlook.com (R.C.); weixianfu@bigc.edu.cn (X.W.); 2Flexible Electronics Research Center, National Institute of Advanced Industrial Science and Technology, Central 5, 1-1-1 Higashi, Tsukuba, Ibaraki 305-8565, Japan; sei-uemura@aist.go.jp (S.U.); t-kamata@aist.go.jp (T.K.); 3Department of Image & Materials Science, Graduate School of Advanced Integration Science, Chiba University, 1-33 Yayoi-cho, Inage-ku, Chiba 263-8522, Japan; nakamura.kazuki@faculty.chiba-u.jp; 4School of Materials Science and Engineering, the Key Laboratory of Material Processing and Mold of Ministry of Education, Henan Key Laboratory of Advanced Nylon Materials and Application, Zhengzhou University, Zhengzhou 450001, China; dcs@gs.zzu.edu.cn (C.D.); liuxy@zzu.edu.cn (X.L.)

**Keywords:** organic nonvolatile transistor memory, polypeptide derivatives, polymer dielectric, molecular alignment

## Abstract

Organic nonvolatile transistor memory with synthetic polypeptide derivatives as dielectric was fabricated by a solution process. When only poly (γ-benzyl-l-glutamate) (PBLG) was used as dielectric, the device did not show obvious hysteresis in transfer curves. However, PBLG blended with PMMA led to a remarkable increase in memory window up to 20 V. The device performance was observed to remarkably depend on the blend ratio. This study suggests the crystal structure and the molecular alignment significantly affect the electrical performance in transistor-type memory devices, thereby provides an alternative to prepare nonvolatile memory with polymer dielectrics.

## 1. Introduction

Organic memory array has attracted extensive research interest in electronic applications, such as in the smart tag, radio frequency identification tag, e-signage, flexible sensor, and flexible display [1,2,3,4,5], due to the advantages in low-temperature processes [6,7,8]. Among the organic memory devices, the ones with thin-film transistor (TFT) configuration are especially attractive because of nondestructive data readout, circuit architectural compatibility, and single transistor features [9,10,11,12]. In the past decades, many attempts have been made to fabricate such a nonvolatile memory device by exploring inexpensive ferroelectric, nano-floating gate dielectric and chargeable polymer-based electrets, which is compatible with printing technique on flexible substrates [13,14,15,16,17,18]. Remarkable progress in flexibility, memory window, and capability of multibit storage have been achieved. 

In a ferroelectric-based TFT memory device, effective controlling the threshold voltage shift on the transfer curves is implemented by tuning the polarization of gate dielectric. The polarization direction in the dielectric was determined by the dipole orientation, thus directly leading to the on and off state of the memory device. Among the various ferroelectric materials, poly (vinylidene fluoride) (PVDF) and its derivative poly(vinylidenefluoride-trifluoroethylene) (P(VDF-TrFE)) have been widely studied, which exhibit large spontaneous polarization and excellent chemical stability and are of particular interest for organic memory device [19]. However, the poor solubility in organic solvent and requiring special treatment such as high-temperature annealing have limited the development in roll to roll mass production. [20] Therefore, developing the solution-processable ferroelectric polymers is necessary for the further development of printed flexible memories. 

In addition, many attempts have been made to realize a further enhancement of the memory characteristics by blending the semiconductors [21] or gate dielectric polymer layer with well-known semiconducting molecules [22,23]. For example, Park et a. constructed the non-volatile transistor memory device through applying semiconductor blends of *p*-channel Tips-pentacene (6,13-bis(triisopropylsilylethynyl) pentacene) and *n*-channel poly{[N,N’-bis(2-octyldodecyl)-naphthalene-1,4,5,8-bis(dicarboximide0-2,6-diyl)]-alt-5,5-(2,2-bithiophene)}(P(NDI2OD-2T); N2200) and polystyrene-brush as the gate electret, they obtained high ON/OFF current ratio about 10^7^ and large memory window of 55 V through optimizing the blends ratio. [21] Chiu et al. reported that the blends of ferroelectric (P(PVDF-TrFE) and n-type semiconducting (PCBM) based memory device showed the high I_ON_/I_OFF_ ratios (≈3 × 10^3^) and excellent retention characteristics up to 10^4^ s [22]. Wang et al. reported that the blending C60 in the PS matrix is a promising method for transistor-based memory devices. [23]. However, the tuning on the FET memory characteristic through blends with ferroelectric and insulator dielectric the have not been fully explored yet. 

In the present paper, based on the previous study [24,25,26], we explored the pentacene-based organic nonvolatile transistor type memory (Figure 1a,b) though using a synthetic polypeptide, poly (γ-benzyl-l-glutamate) (PBLG), bearing benzyl group as an end group of the side chain (Figure 1c), and a reference poly (γ-methyl-l-glutamate) (PMLG) with higher degree of polymerization (185) as the dielectric (Figure 1d), respectively. Both of these two ferroelectrics 24–26 have highly ordered α helix structure, in which hydrogen bonds are formed parallel to the direction of the molecular axis as shown in Figure 1e. The physical and chemical characteristics of dielectric polymers were studied. In addition, a further enhancement of the memory window by blending the ferroelectric polymer with defect-free material of PMMA was investigated. Our experimental results provide a systematic study on establishing the relationship between the memory characteristics and the polymer blend composition and also provide a simple route for the development of non-volatile transistor memory devices. 

## 2. Experiment Section

### 2.1. Materials

The PMLG (degree of polymerization: 185) and PBLG (degree of polymerization: 180) were provided by Kyowa Hakko Kogyo Co. Ltd. (Tokyo, Japan). Pentacene (98% purity) was provided by Nard Institute Ltd. (Tokyo, Japan) and was used as received without further purification. 1,2-Dichloroethane was provided by Kanto Kagaku Co., Inc. (Tokyo, Japan). Other chemicals were purchased from Tokyo Chemical Industry Co., Ltd. (Tokyo, Japan).

### 2.2. The Fabrication of TFT Memory

TFT memory devices were fabricated by depositing a pentacene layer (film thickness = 50 nm) as an active layer at a pressure of 2 × 10^−3^ Pa and an evaporation rate of 0.2–0.4 Å s^−1^ on the ITO/PMLG or ITO/PBLG film. Au as the source and drain electrodes (W/L = 5 mm/20 μm) was deposited by vacuum evaporation on pentacene film. The TFT structure using a top contact and gate bottom geometry is schematically depicted in Figure 1 with the chemical structures of polypeptide complex and pentacene. 

### 2.3. Apparatus

All electric measurements were carried out using a Keithley 4200 semiconductor parameter analyzer (California, USA) under dark conditions in vacuo. The device for the thermally stimulated depolarization current (TSDC) measurement was heated by a thermo-controller (pREX100, Rikendensi, Japan), and the depolarized currents were measured by a micro-amperometer (617, Keithley, California, USA). X-ray diffraction (XRD) experiments were conducted using UltraX 18SF (Rigaku, Japan), which employs a Cu Kα_1_ Source with a wavelength of 0.154 nm (40 kV, 200 mA). The optical micrographs were taken by a digital microscope (Keyence Co., Ltd., Japan, VHX-2000).

## 3. Results and Discussion

The surface structure of pure PMLG and PBLG film prepared with the spin coating process was evaluated by the atomic force microscopy (AFM) images as shown in Figure 2. The surface morphology of PBLG film featured fibril-like microstructures, while the PMLG film shows the granules structures with the size of several tens of nanometers. Additionally, we examined the surface morphology of PMLG (degree of polymerization: 185) that was also prepared with spin-coating process shown in Appendix A. The film also shows the network of distinguished long fibril structures, indicating that compared with the methyl group, the higher degree of polymerization and dense packing pattern of benzyl group alignment in the side chain leads to the strong intermolecular interaction in the main chain. 

In addition, UV/Vis absorption spectra of both polypeptide films (the thickness of PMLG and PBLG film are 800 nm and 200 nm, and prepared with blade coating and spin coating, respectively) and annealed PBLG film were collected in Figure 3. Both the polypeptide films exhibit a specific absorption band from 180 nm to 260 nm with the same maximum absorption at 190 nm, which was assigned as the characteristics absorption of the π−π∗ stacking carbonyl groups alignment in the main chain of the polypeptide. However, in the case of PMLG film, another shoulder peak around 210 nm which could be due to the n−π∗ absorption also was obtained. At the same time, in order to probe the conformation of these two polypeptides film, the CD spectral has also been investigated. As shown in Figure 3a,b, both the PMLG and PBLG film shown the positive and negative CD signals. So it is evident that both the PMLG and PBLG possess the α helix structure. However, a positive Cotton effect at about 205 nm and a negative Cotton effect at 210 nm film only can be observed in PMLG, which was not obtained in PBLG indicating that PBLG film prepared with spin-coating process does not possess the cholesteric liquid crystalline structure. In addition, from Figure 3c, the high-temperature annealing of PBLG film yielded a slight absorption intensity decrease in the UV spectral, however, there were no significant attenuated CD signals, indicating the good thermal stability and structural regularity in the obtained PBLG polypeptide film.

In order to further investigate the different microstructure of both the PMLG and PBLG film, the wide-angle X-ray diffraction (WAXD) was performed as exhibited in Figure 4a,b. In particular, a broad diffraction peak at a 2θ of 7~9° from the PMLG film was obtained. However, in the case of the PBLG film, only a sharp reflection peak at 2θ of 7° was obtained, indicating that compared with PMLG film, PBLG film forms larger crystal grain and has higher crystallinity. Appendix A also exhibits the XRD spectra of the PMLG (degree of polymerization: 185) film, where a sharp diffraction peak at 8° also could be observed, indicating that the sharp reflection peak of PBLG film appearing in the small-angle region was presumably due to the compacted packing pattern of the side chain benzyl dipoles as confirmed by the AFM images above. In addition, Appendix A shows the DSC curves of the PBLG film at a temperature range from −50 to 150 °C on the heating runs. The results further confirmed that the PBLG film shows the high crystallinity, indicating the side chain relaxation and movement are restricted within this temperature range.

Implementation of these two polypeptides as the dielectric in electronic devices was evaluated in transistor memory. Figure 5a–d show the transfer and output characteristics using PMLG or PBLG dielectric, respectively. It is clearly observed that these transfer curves exhibit a typical p-type accumulation mode. Moreover, the PBLG-based device had a lower threshold voltage than that of the PMLG-based device. The ON/ OFF ratio of PBLG based device was 10^4^, which is the same value as PMLG-based memory device. Additionally, the observed turn-on voltage of PBLG-based device was also lower than that of PMLG-based transistor device. The remarkable difference was that the transfer characteristic of the ferroelectric PMLG-based memory device showed a pronounced memory window, however, the hysteresis behavior was not observed in PBLG-based transistor device. Therefore, it is very interesting to study the memory mechanism of PBLG-based device in detail. Both of these two polypeptides possess the same α helix structure as discussed above, which composes of the main chain dipole originating from the amide bonds and the side chain dipole [24,25].

In order to study the dipole movement which plays an important role in the ferroelectric behavior of the polypeptide-based memory device, the TSDC spectra of ITO/polypeptide (film thickness: 1 μm)/Au device has been measured. Figure 6 shows the TSDC curves of polypeptide-based MIM devices with temperature range from (a) −40 to 100 °C and (b) −60 to 100 °C, the MIM device was polarized at Tp = 80 °C (Figure 6a) and Tp = 50 °C (Figure 6b) by applying voltage of 10 V for 5 min, respectively. Then the device was short–circuited for 5 min to remove excess charges which accumulated at the interface between electrode and dielectric layer. Further, the TSDC current was measured by applying the collected voltage of 0.5 V at a constant heating rate of 6.6 °C. As shown in Figure 6a, interestingly, two clearly negative peaks at both 50 °C and 90 °C for PBLG and PMLG films were observed, respectively. It is notable that the phase transition temperature of the α-helix main chain of polypeptide was 80 °C. Therefore, the depolarization electric current peak at 90 °C could be attributed to the polarization relaxation of the dipole moment originating from the amide group. On the other hand, when the two devices were polarized at Tp = 50 °C, the dielectric relaxations were different substantially as shown in Figure 5b. A clearly single negative peak at around 50 °C was observed when the PBLG film was selected as the dielectric, in which the same peak also observed at the Tp = 80 °C (Figure 6a), indicating that the depolarized peak was attributed to the motion of side-chain dipole of the PBLG polypeptide, namely, the dipole in the side chain could be oriented through applying the electric field at a temperature higher than 50 °C as illustrated in Figure 6c. Therefore, it is evident that there is no memory window exhibiting in the transfer characteristic at room temperature (Figure 5d). However, in the case of PMLG film, as shown in Figure 6b, a negative depolarization peak around 70 °C was observed, which could be ascribed to the phase transition of the ferroelectric relaxation of the PMLG film.

In order to study how the packaging structure of the side chain of PBLG affects the memory function, the PBLG/PMMA blends with different PMMA concentration were prepared as the dielectric in the memory device. Figure 7 shows the transfer characteristics of memory devices with different PBLG/PMMA blends. There is no memory window in the PBLG/PMMA (9:1 and 4:6) device. However, surprisingly, a large window loop up to 20V was obtained with a moderate blend ratio (8:2 and 6:4), indicating that the blend concentration of PMMA plays an important role in the electronic performance of the non-volatile transistor memories. Further, it is interesting to observe that the memory loop shrank when further increasing the concentration of PMMA, which could be attributed to the nanophase separation of the PBLG/PMMA blends film as illustrated in the AFM analyses (Figure 8). In Figure 2b, a continuous network of PBLG fibrils can be observed for the pure PBLG film, but two distinct phases can be observed in the AFM images of the blend films with the increasing of the PMMA content (Figure 8a–d), and the PMMA phase almost covered on the surface of the PMLG film (6:4). These results are consistent well with the XRD characteristics. Figure 8e shows the XRD patterns of pure PBLG and the polymer blends with PMMA. With less PMMA in the blend, there is a remarkable decrease in the peak intensity and increase in the full-width-half-maximum (FWHM) of the XRD peaks, and the specific value was summarized in Appendix A, indicating that the ordered crystalline structure of PBLG molecules was damaged.

In consequence, in the case of the pristine-PBLG-based device, the dipoles in the ferroelectric phase of PBLG molecules are difficult to be rotated and aligned with the variation of electric field, so there was no clear memory window depicted in Figure 5d. Instead, with the concentration of the PMMA increasing, because of the phase separation, the PMMA molecules could intercalate into the PBLG molecules, so the polarization of the side chain dipole was enhanced with low crystallinity (Appendix A), leading to memory behavior. But the strongly isolated cluster-like features of the PBLG phase can be seen in the composition (4:6) of PBLG/ PMMA blend film (As shown in Figure 8d), which could be ascribed to the agglutination of PBLG molecules. Therefore, the motion of side-chain was restrained, thus the memory loop disappeared. Our results provide evidence that an appropriate blend of two polymers could change the crystallinity of the polypeptide dielectric, leading to the memory function in the organic thin-film transistor device. This provides a possible route to fabricate memory devices through controlling the crystal patterning in the ferroelectric gate dielectric.

## 4. Conclusions

In summary, we have fabricated an organic nonvolatile transistor type memory with synthetic polypeptide derivatives, and studied the dipole movement of the ferroelectric PBLG film though applying the TSDC technique, and achieved the memory device when the appropriate phase-separation blends of PBLG and PMMA. Our devices demonstrated excellent electrical performance with a large memory window (20 V), high ON/OFF ratio (≈10^4^) through appropriate phase separation. This work opens a simple way to prepare the memory devices, and could be broadly extended to many other ferroelectric gate dielectrics.

## Figures and Tables

**Figure 1 molecules-25-00499-f001:**
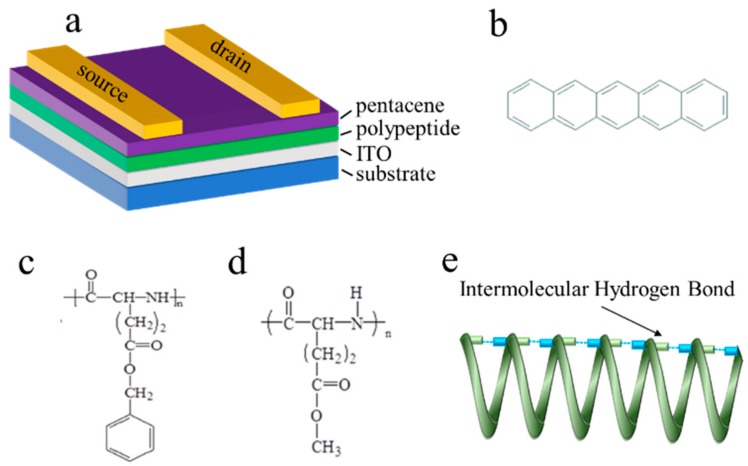
(**a**) The configuration of bottom-gate top-contact BioTFT device with thermally deposited Au as source (S) and drain (D) electrodes, polypeptide as the gate dielectric layer, pentacene is used as the semiconductor, and Indium tin oxide (ITO) as the gate electrode. The chemical structures of (**b**) pentacene, (**c**) poly (γ-benzyl-l-glutamate) (PBLG) and (**d**) poly (γ-methyl-l-glutamate) (PMLG). (**e**) The α helix structure of polypeptide polymer.

**Figure 2 molecules-25-00499-f002:**
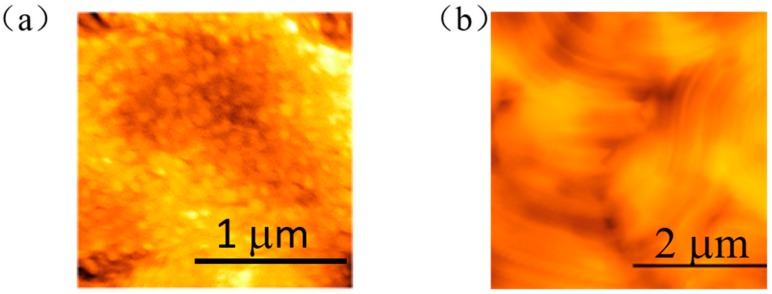
Top-surface atomic force microscopy (AFM) images of the deposited (**a**) PMLG and (**b**) PBLG film, respectively.

**Figure 3 molecules-25-00499-f003:**
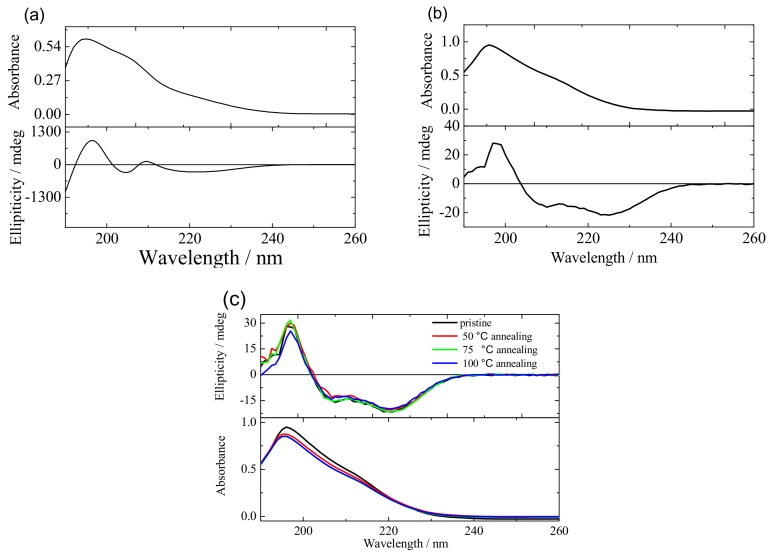
UV absorption and circular dichroism (CD) spectra of (**a**) as-deposited PMLG, (**b**) as-deposited PBLG film and (**c**) annealed PBLG film.

**Figure 4 molecules-25-00499-f004:**
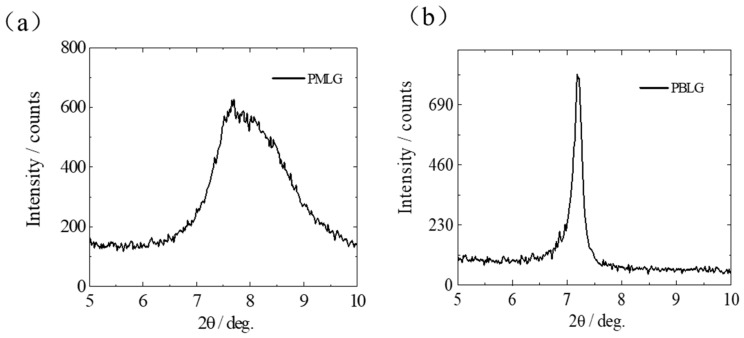
X-ray diffraction (XRD) curves of the (**a**) PMLG (**b**) PBLG polypeptide thin films.

**Figure 5 molecules-25-00499-f005:**
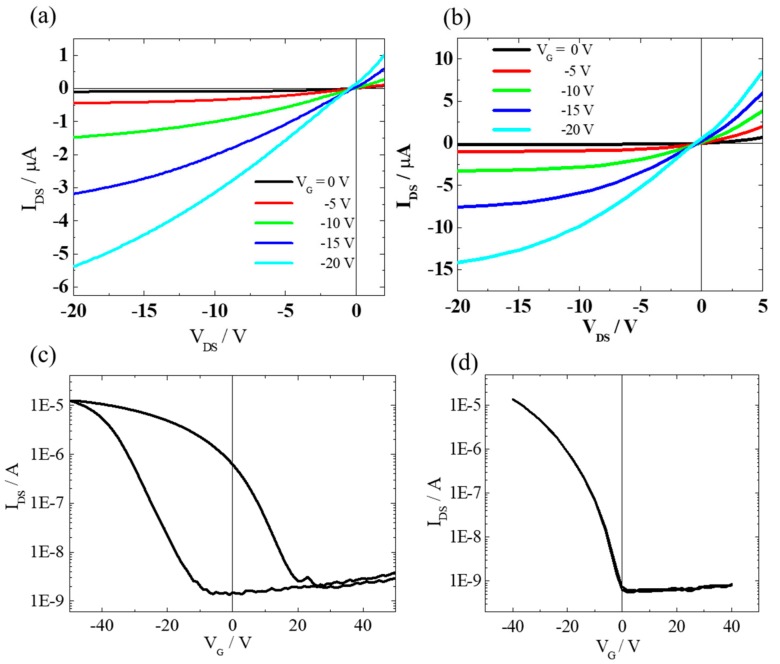
The electrical performance of organic TFT (OTFT) memory device with polypeptide gate dielectric. The output characteristic of drain-source current versus drain-source voltage (I_DS_–V_DS_) plot of (**a**)PMLG (**b**) PBLG based device. The transfer characteristic of drain-source current versus gate voltage (I_DS_ vs. V_G_) plot of (**c**) PMLG (**d**) PBLG based device. (**c**) PMLG (**d**) PBLG based device.

**Figure 6 molecules-25-00499-f006:**
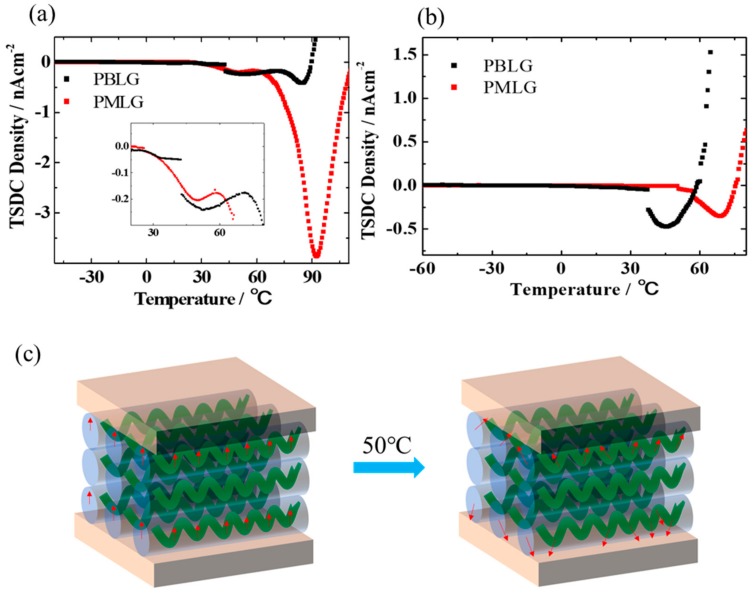
Thermally stimulated depolarization current (TSDC) spectra of ITO/PMLG/Au and ITO/PBLG/Au device at (**a**) 80 °C, (**b**) 50 °C. The insert in (**a**) shows the curves from 20 to 80 °C. (**c**) Illustration of the depolarization process of PBLG.

**Figure 7 molecules-25-00499-f007:**
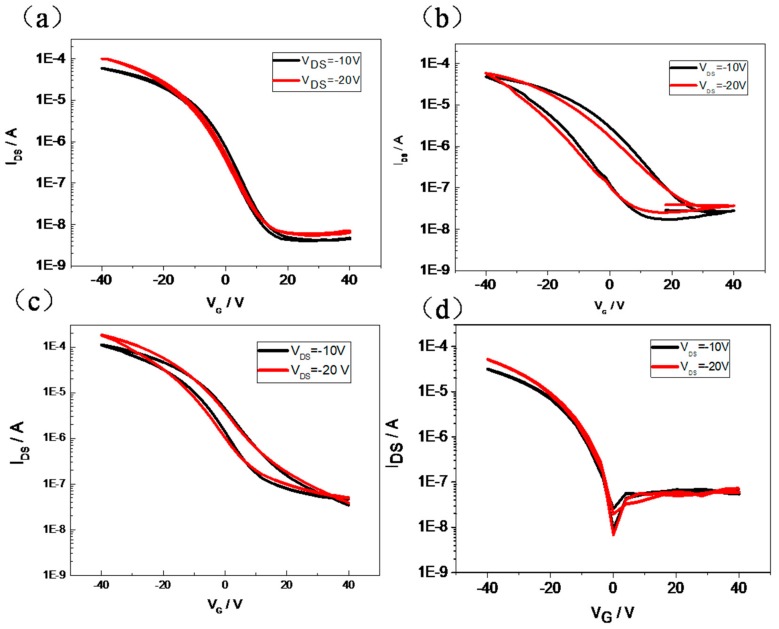
Transfer curves at V_d_ = −10V and −20 V of transistor memory devices with PBLG/PMMA blends as gate dielectric. The PBLG/PMMA ratios are (**a**) 9:1, (**b**) 8:2, (**c**) 6:4, (**d**) 4:6.

**Figure 8 molecules-25-00499-f008:**
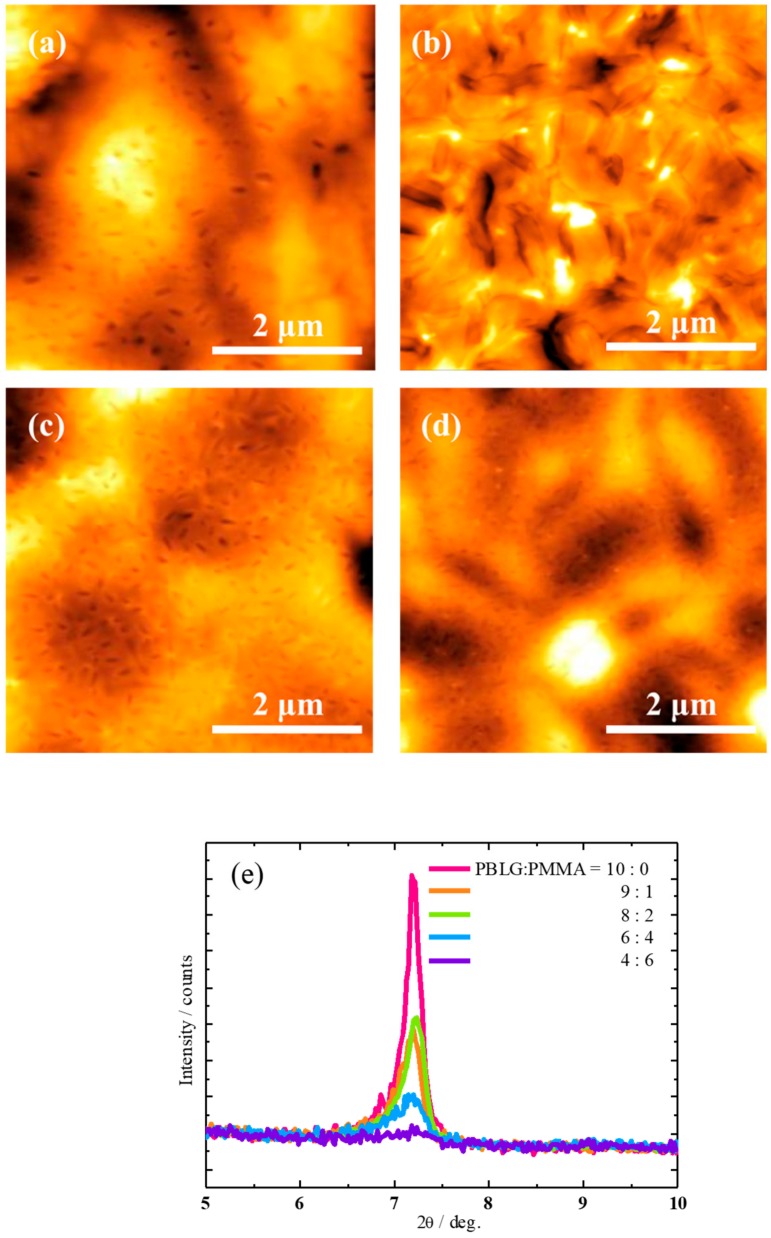
(**a**–**d**) AFM images of PBLG/PMMA films with different PBLG/PMMA ratios spun on ITO substrates. The PBLG/ PMMA ratios are (**a**) 9:1, (**b**) 8:2, (**c**) 6:4, (**d**) 4:6. In all AFM images, the scale bar presents 2 μm. (**e**) X-ray diffraction patterns of the blend films.

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
