# Peer review of "Non-Volatile Transistor Memory with a Polypeptide Dielectric"

_molecules, 2020, doi:10.3390/molecules25030499_

Round 1
Reviewer 1 Report
In this study, poly (γ-benzyl-L-glutamate) (PBLG) and PMMA bled based FET has been shown memory window up to 20 V due to its ferroelectric characteristics. However, the there are several misleading description and unclear mechanism. Hence, I would suggest acceptance after major revision with some critical remarks. The following are the missing parts that should be further clarified.
In line 92 of page 3, the full name of TSDC should be provided. In Figure 2, the scale range and sequence of Circular Dichroism (CD) and UV absorption spectra should keep consistent in the three sample. In the caption of Figure 4, “(b) The transfer characteristic of drain –source current versus gate voltage” where (b) should be removed. The author have claimed the ferroelectric characteristics of PBLG , PMLG and PMMA are the main factor to control the memory effect. However, the threshold shift may result from trapped charge by dielectric layer. The ferroelectric hysteresis loops should be measure or the relevant references should be provided to claim the ferroelectric characteristics of the studied polymers.
Author Response
Response:
In line 92 of page 3, the full name of TSDC was added. Please see “ The device for the thermally stimulated depolarization current (TSDC) measurement……” In Figure 2, the scale range and sequence of Circular Dichroism (CD) and UV absorption spectra were revised. In the caption of Figure 4, “(b) The transfer characteristic of drain –source current versus gate voltage” where (b) should be removed. According to our previous study, this threshold voltage shift was not caused by trap sites but the ferroelectric character, And our results reveal that the window loop was remarkably related to the crystallinity in PBLG.

Reviewer 2 Report
The major novelty of this work refers to develop a PBLG/PMMA hybrid as dielectric for organic nonvolatile transistor. The transistor performance depends on the PBLG/PMMA ratio. However, the manuscript is clearly written in a hurry with plenty of typing and language errors. Also the scientific information provided is very chaotic and contains a large number of assumptions. Exactly here there are a number of questions that remain unanswered or that could be improved.
The major story in this manuscript is focused on the PBLG and the PBLG/PMMA. However, the authors took a great attention on the PMLG, which diluted the whole story and confused the readers. In addition, the PMLG as dielectric for transistor has been heavily reported even by the authors themselves. The paper structure is suggested to be re-organized. The authors draw an important conclusion about the role of blending ratio of PBLG/PMMA on the transport performance. However, the relevant mechanism explanation is too weak. What is the influence of the film thickness of the PBLG/PMMA hybrid? How about the related film crystallinity, stacking orientation and surface roughness. Other small issues for example: 1) The authors listed the references on the previous study [24-26]. Where is the reference 26? 2) The polymerization degree for different polypeptide is confusing. PMLG 185 or 180 or 440? 3) For the UV-Vis curves, it is quite odd to mark the wavelength in X axis as 207, 253 nm. 4) For the XRD curve, how about the patterns between 10-40 degree?Author Response
Response:
We added the mechanism explanation in Figure 6 and Figure S4. And also, the description was modified. The influence of film thickness of the hybrid is rather weak if there is no remarkable phase separation. According to our observation, PBLG and PMMA can be well mixed with nanoscale phase separation. The reference of 26 was added. “a reference PMLG with higher degree of polymerization (185) as the dielectric”. The information was added. The sentence was revised. Please see “Both the polypeptide films exhibit specific absorption band from 180 nm to 260 nm with the same maximum absorption at 190 nm,”. XRD was used to determine the molecular packing. Therefore, the peaks appearing around 10 are necessary. As described, “the sharp reflection peak of PBLG film appearing in the small-angle region was presumably due to the compacted packing pattern of the side chain benzyl dipoles “

Round 2
Reviewer 2 Report
The authors put effort to further solidify the work. I would like to recommend its publication now.